# Cryo-EM structures of a pentameric ligand-gated ion channel in liposomes

**Vikram Dalal, Brandon K Tan, Hanrui Xu, Wayland WL Cheng\***

Department of Anesthesiology, Washington University School of Medicine, St Louis, United States

## eLife Assessment

The reported cryo-EM imaging of a pentameric ligand-gated ion channel in liposomes as opposed to nanodiscs has both broad implications and contributes **valuable** methodological advances to the structural investigation of membrane receptors. The comparison of structures assigned to distinct functional states in liposomes versus nanodiscs is **convincing** and will aid membrane protein structural biologists in selection of functionally relevant membrane reconstitution environments.

**\*For correspondence:**
wayland.cheng@wustl.edu

**Abstract** Detergents and lipid nanodiscs affect the cryo-EM structures of pentameric ligand-gated ion channels (pLGICs) including ELIC. To determine the structure of a pLGIC in a membrane environment that supports ion channel function, we performed single particle cryo-EM of ELIC in liposomes. ELIC activation and desensitization were confirmed in liposomes with a stopped-flow thallium flux assay. Using WT ELIC and a non-desensitizing mutant (ELIC5), we captured resting, activated, and desensitized structures at high resolution. In the desensitized structure, the ion conduction pore has a constriction at the 9' leucine of the pore-lining M2 helix, indicating that 9' is the desensitization gate in ELIC. The agonist-bound structures of ELIC in liposomes are distinct from those in nanodiscs. In general, the transmembrane domain is more loosely packed in liposomes compared to nanodiscs. It has been suggested that large nanodiscs are superior for supporting membrane protein function. However, ELIC localizes to the rim of large circularized nanodiscs, and structures of ELIC in large nanodiscs deviate from the liposome structures more than those in small nanodiscs. Using liposomes for cryo-EM structure determination of a pLGIC increases our confidence that the structures are snapshots of functional states.

## Introduction

There are many cryo-EM structures of pentameric ligand-gated ion channels (pLGICs) that have provided insight into the architecture of these ion channels and the determinants of activation and desensitization. Ideally, structure determination should be performed under conditions in which ion channel function (i.e. ion flux or conduction) can also be measured. However, all available high-resolution structures of pLGICs have used proteins solubilized in detergent or reconstituted in one of a variety of lipid nanoparticles or nanodiscs. In many cases, the structures vary depending on the detergent or nanodiscs used. In the α1 glycine receptor (GlyR), the proportion of particles contributing to open and desensitized conformations differed when using detergent, MSP (membrane scaffold protein) nanodiscs, or SMA (styrene maleic acid) extracted nanoparticles (*Yu et al., 2021*). In other pLGICs, different structures were obtained in detergent or various nanodiscs for the GABA$_A$ receptor (GABA$_A$R) (*Zhu et al., 2018*; *Kim et al., 2020*), serotonin 3A receptor (5-HT3aR) (*Basak et al., 2018a*; *Basak et al., 2018b*; *Zhang et al., 2021a*), α7 nicotinic acetylcholine receptor (α7 nAChR) (*Noviello et al., 2021*; *Burke et al., 2024*; *Zhao et al., 2021*), and muscle nAChR (*Rahman*

*et al., 2020*; *Zarkadas et al., 2022*). In a systematic study of the prokaryotic pLGIC, ELIC, in which various lipid nanoparticles or nanodiscs were tested, it was shown that the choice of nanodisc affects the agonist-bound structure of ELIC (*Dalal et al., 2024*). Taken together, these studies indicate that detergent or nanodisc conditions impact the structure of pLGICs. Without knowledge of ion channel function under these conditions, the functional annotation of these structures is less certain.

It has been suggested that large nanodiscs, which have proportionally more lipid at the core than rim (the 1–2 nm shell adjacent to the scaffold), may be superior for structural studies of pLGICs. This is because molecular dynamics simulations of ELIC in small nanodiscs show frequent contact of the scaffold with the membrane-facing M4 helix, altering the conformation and dynamics of this helix (*Dalal et al., 2024*). The MSP also perturbs the structure of the lipid bilayer (i.e. thickness and order) at the rim (*Bengtsen et al., 2020*), and this too may be undesirable. Therefore, it is possible that larger nanodiscs are better membrane mimetics, reducing interactions of the membrane protein with the scaffold or having more homogeneous lipid bilayer properties. In support of this notion, the function of the ABC transporter was found to be sensitive to nanodisc size with larger nanodiscs yielding functional properties that are more similar to the ABC transporter in liposomes (*Nouel Barreto et al., 2025*).

Here, we report cryo-EM structures of ELIC in liposomes where the ion channel is confirmed to be in resting, activated, and desensitized states. By performing structural and functional characterization under the same reconstitution conditions, we increase our confidence in the functional annotation of these structures. A comparison of structures in liposomes and nanodiscs shows differences in the packing of the transmembrane domain (TMD) and M4 transmembrane helix. Contrary to what was suggested previously (*Dalal et al., 2024*), structures of ELIC in large nanodiscs deviate the most from those in liposomes.

## Results

### Resting, activated, and desensitized structures of ELIC in liposomes

Optimal reconstitution of ELIC in liposomes to facilitate single particle cryo-EM was achieved by dialysis. Reconstitution of ELIC in liposomes at low protein:lipid molar ratios can be accomplished by detergent removal with Biobeads (*Petroff et al., 2022*); however, reconstitution at a higher protein:lipid molar ratio (1:500 in this study) with efficient incorporation of ELIC in the liposomes required dialysis. The method was adapted from protocols previously reported for the reconstitution of muscle nAChR (*daCosta et al., 2009*), 5-HT3aR (*Kudryashev et al., 2016*), and ELIC (*Carswell et al., 2015*). Although dialysis was performed for 5–7 days to ensure complete removal of detergent, the minimum time of dialysis necessary was not tested. A shorter dialysis protocol could likely be implemented for less stable proteins depending on the detergent and dialysis conditions (*Zumbuehl and Weder, 1981*). Proteoliposomes were formed using a 2:1:1 molar ratio of POPC (1-palmitoyl-2-oleoyl-phosphatidylcholine):POPE (1-palmitoyl-2-oleoyl-phosphatidylethanolamine):POPG (1-palmitoyl-2-oleoyl-phosphatidylglycerol); these lipids are known to support ELIC activity (*Petroff et al., 2022*), and phosphatidylethanolamine and phosphatidylglycerol are the two principal phospholipids found in gram-negative bacteria in which ELIC is expressed. Using a stopped-flow thallium flux assay, we confirmed that WT ELIC activates and then mostly desensitizes in response to the agonist, propylamine (*Figure 1A*). The time constant of activation and desensitization was 10 ± 2 ms and 5.1 ± 2.1 s (± SD, $n$ = 6), respectively, which matches the kinetics of ELIC activation and desensitization from excised patch-clamp recordings in liposomes or HEK293 cells (*Marabelli et al., 2015*; *Laha et al., 2013*; *Tong et al., 2019*). We previously identified a combination of five mutations in ELIC, P254G/C300S/V261Y/G319F/I320F, called ELIC5 which eliminates channel desensitization (*Petroff et al., 2022*). We confirmed that ELIC5 in 2:1:1 POPC:POPE:POPG liposomes activates and does not desensitize for at least 20 min (*Figure 1A*). The open probability of ELIC5 when activated by 10 mM propylamine in 2:1:1 POPC:POPE:POPG membranes is estimated to be close to 1 (*Petroff et al., 2022*). Therefore, proteoliposome preparations of WT ELIC in the absence and presence of propylamine are expected to yield resting and desensitized channels, respectively, and ELIC5 in the presence of propylamine is expected to yield activated channels. The same reconstitution protocol was used for the stopped-flow thallium flux assay and single particle cryo-EM.

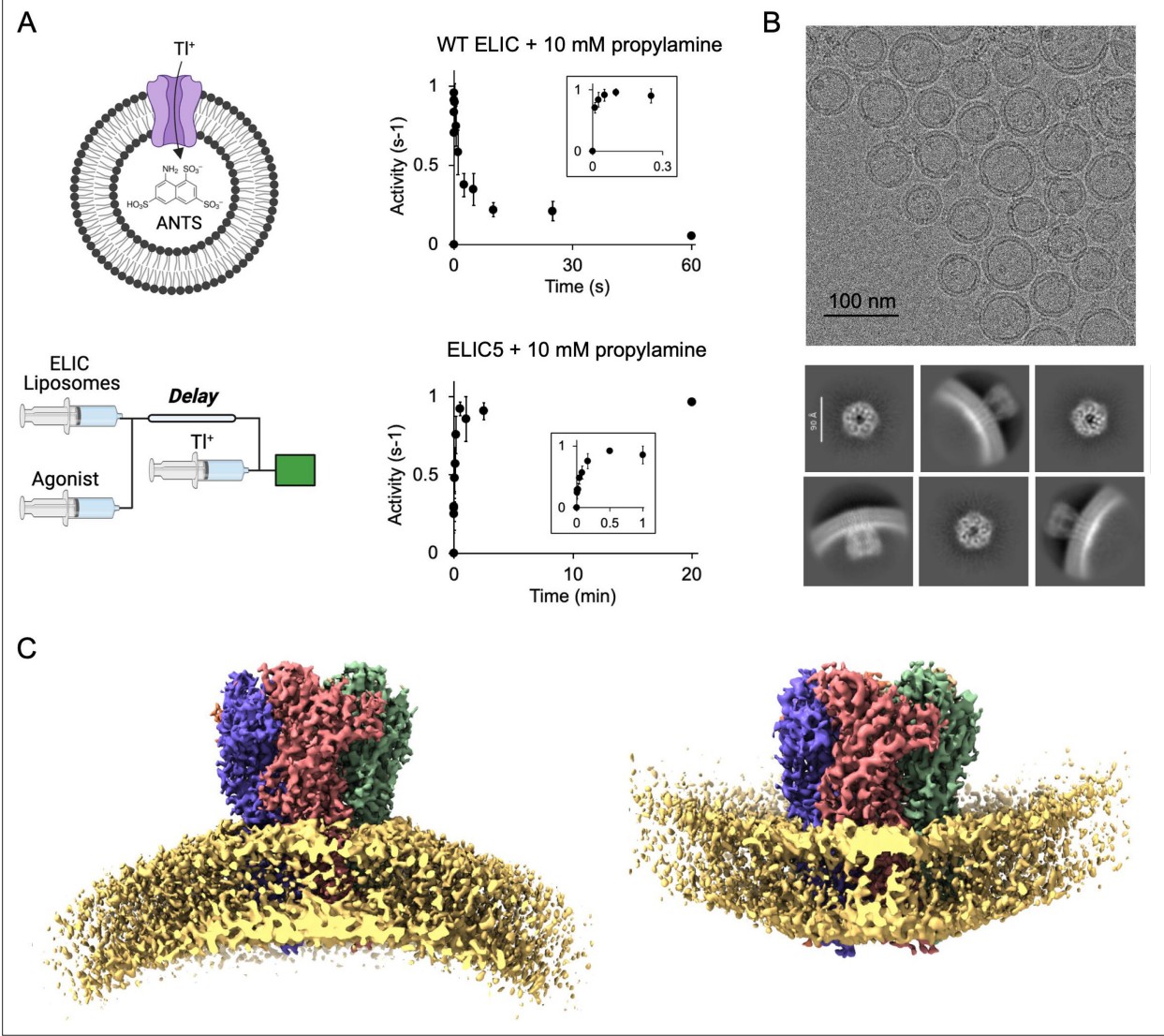

**Figure 1.** Single particle cryo-EM of ELIC in liposomes. (**A**) Schematic of the fluorescence stopped-flow liposomal flux assay: a sequential mixing experiment is performed. ELIC proteoliposomes are mixed with agonist followed by thallium (Tl⁺) after a variable delay time. ELIC activity at each delay time is measured as the rate of fluorescence quenching by the influx of Tl⁺. Shown is WT ELIC and ELIC5 activity after mixing with 10 mM propylamine over varying delay times ($n$ = 3–6, ± SEM). (**B**) Representative cryo-EM micrograph of WT ELIC in 2:1:1 POPC:POPE:POPG liposomes and 2D class averages. (**C**) Cryo-EM maps of WT ELIC without agonist in liposomes showing inward- and outward-facing orientations. The lipid bilayer density is shown at a lower contour level for illustration.

The online version of this article includes the following figure supplement(s) for figure 1:

**Figure supplement 1.** Summary of single particle cryo-EM analysis of WT ELIC in liposomes.

**Figure supplement 2.** Summary of single particle cryo-EM analysis of ELIC5 with agonist in liposomes, similar to *Figure 1—figure supplement 1*.

**Figure supplement 3.** Summary of single particle cryo-EM analysis of WT ELIC with agonist in liposomes, similar to *Figure 1—figure supplement 1*.

**Figure supplement 4.** Display of local resolution on sharpened maps and FSC curves from outward- and inward-facing ELIC structures in liposomes.

We determined structures of WT ELIC with and without agonist, and ELIC5 with agonist. Side views from the 2D class averages showed that ELIC is oriented in both directions in the liposomes with a preference for the extracellular domain (ECD) facing the extra-liposomal solution (i.e. outward-facing orientation) (*Figure 1B*, *Figure 1—figure supplement 1*, *Figure 1—figure supplement 2*, *Figure 1—figure supplement 3*). Heterogeneous refinement in CryoSPARC separated the particles into two classes that generated outward- and inward-facing 3D reconstructions (*Figure 1C*). One caveat to this analysis is that top/down views of outward- and inward-facing channels may not be distinguishable

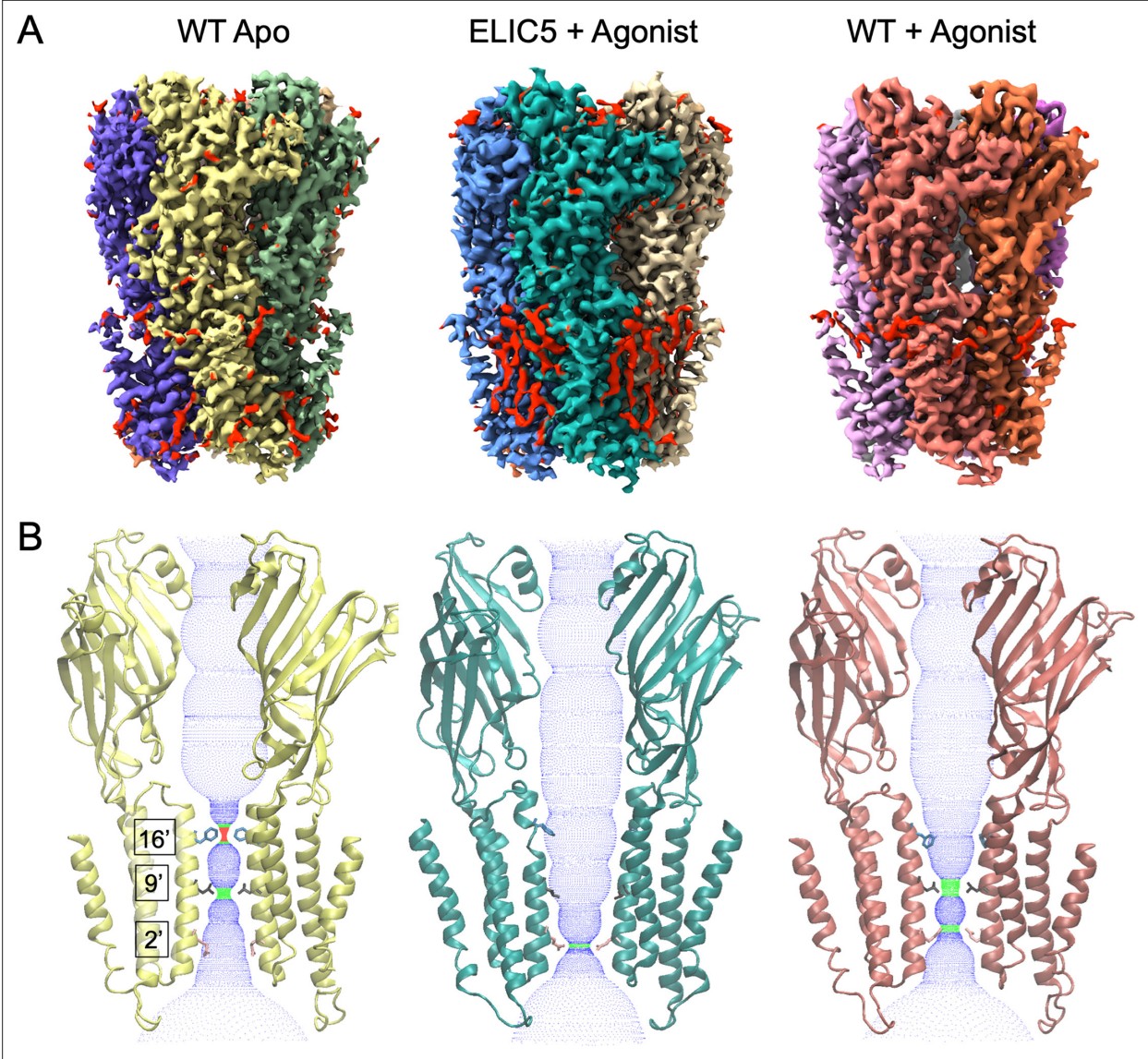

**Figure 2.** Structure and pore conformation of resting, activated, and desensitized states. (**A**) Cryo-EM maps of WT ELIC without agonist (resting state), ELIC5 with 10 mM propylamine (activated state), and WT ELIC with 10 mM propylamine (desensitized state) in liposomes. In the ELIC5 + Agonist map, red shows non-protein densities. (**B**) Two opposing subunits from models of the structures shown in (**A**) along with a dot plot of the pore dimensions generated using HOLE (**Smart et al., 1996**). Shown are the side chains for F247 (16'), L240 (9'), and Q233 (2').

The online version of this article includes the following figure supplement(s) for figure 2:

**Figure supplement 1.** Comparison of resting (WT Apo), activated (ELIC5 + Agonist), and desensitized (WT + Agonist) structures of ELIC in liposomes.

**Figure supplement 2.** Pore structure of ELIC in liposomes.

**Figure supplement 3.** Cryo-EM map of ELIC5 with agonist in liposomes with non-protein lipid-like densities colored in red.

without the curvature of the liposome membrane as a reference. Therefore, it is possible that outward- and inward-facing ELIC particles could not be rigorously separated using 3D heterogeneous refinement or 3D classification in CryoSPARC or Relion. Accordingly, the proportion of particles assigned to outward- and inward-facing 3D classes may not accurately reflect the relative orientation of ELIC in the liposomes. Nevertheless, final maps were obtained for apo WT ELIC at a resolution of 3.4 Å for outward-facing and 3.6 Å for inward-facing channels, agonist-bound WT ELIC at a resolution of 3.8 Å for outward-facing and 4.2 Å for inward-facing channels, and agonist-bound ELIC5 at a resolution of 2.8 Å for outward-facing and 4.2 Å for inward-facing channels (**Figure 2A**, **Figure 1—figure supplement 1**, **Figure 1—figure supplement 2**, **Figure 1—figure supplement 3**, **Figure 1—figure**

*supplement 4*, *Supplementary file 1*). In all samples, the inward- and outward-facing structures of ELIC were not significantly different. However, we cannot be certain that ELIC structure is insensitive to membrane curvature due to the limitations of distinguishing inward- and outward-facing particles. In addition, the inward-facing structures were of significantly lower resolution and quality compared to the outward-facing structures. The reason for this is uncertain, but could be due to fewer inward-facing particles, increased structural heterogeneity, and contamination within the liposomes. We focused our analysis on the outward-facing structures since these have better resolution. Moreover, only outward-facing channels are activated by agonist in the stopped-flow thallium flux assay.

The structures of ELIC in liposomes have the same overall architecture as structures of ELIC or other pLGICs in nanodiscs, consisting of a homopentameric assembly around a central ion conduction pore (*Figure 2*). Each subunit consists of an ECD and TMD. In the agonist-bound structures, propyl-amine binds in a conserved site between β-sheets from two adjacent subunits. Comparison of the apo WT ELIC and agonist-bound ELIC5 structures shows the conformational changes for activation, and comparison of agonist-bound ELIC5 and agonist-bound WT ELIC structures shows the conformation changes for desensitization. ELIC activation consists of a contraction of the agonist binding site, counter-clockwise twisting of the ECD, and outward translation of M2–M3 linker and pre-M1 linker in the TMD (*Figure 2—figure supplement 1*). These changes are also associated with an outward translation and counter-clockwise twisting of the transmembrane helices (M1–M4) that leads to opening of the ion channel pore lined by the M2 helix (*Figure 2—figure supplement 1*). ELIC desensitization consists of a partial return of the transmembrane helices (clockwise twisting and contraction), but without significant change in the ECD or the M2–M3 and pre-M1 linkers in the TMD (*Figure 2—figure supplement 1*).

The structure of the pore also reveals the conformational state of ELIC: the presence or absence of a continuous and sufficiently wide water-accessible pathway indicates whether the ion channel is open or closed, respectively. Apo WT ELIC shows a pore with two hydrophobic constrictions at F247 and L240 (also known as 16' and 9' positions of M2), where the ion conduction pathway is dehydrated and therefore, closed (*Figure 2B*, *Figure 2—figure supplement 2*). In contrast, the agonist-bound ELIC5 pore is hydrated and open because M2 rotates and F247 and L240 orient away from the pore axis toward the adjacent subunit (*Figure 2B*, *Figure 2—figure supplement 2*). There is some narrowing of the pore at 2' (Q233, diameter of 4.4 Å), but this hydrophilic side chain, which can take on different rotameric states (*Petroff et al., 2022*), is not anticipated to impede ion conduction. Agonist-bound WT ELIC shows conformational heterogeneity at F247, and two states were modeled with different conformations for F247 and Y248 (*Figure 2—figure supplement 2*). In State 1, F247 orients toward the pore axis, while in State 2, F247 orients toward M2 of the adjacent subunit. The latter has a wider pore at 16' (F247), but still 9' (L240) forms a hydrophobic constriction (diameter of 3.3 Å) like Apo WT ELIC (diameter of 2.6 Å) (*Figure 2B*, *Figure 2—figure supplement 2*). There is also a narrowing of the pore at 2' (Q233) in the agonist-bound WT ELIC structure (diameter of 3.7 Å). Nevertheless, the pore diameter at 9' (L240) and the hydrophobicity of this residue suggest that 9' is the highest barrier to ion conduction in the agonist-bound WT ELIC structure. In summary, apo and agonist-bound WT ELIC are closed structures with hydrophobic constrictions at 9' consistent with resting and desensitized states, and agonist-bound ELIC5 is an open structure consistent with an activated state. It has been proposed that the activation and desensitization gates in the GABA$_A$R and GlyR are distinct with 9' being the activation gate and –2' the desensitization gate (*Yu et al., 2021*; *Masiulis et al., 2019*; *Kumar et al., 2020a*; *Gielen and Corringer, 2018*; *Gielen et al., 2015*). However, in agonist-bound structures of the α7 nAChR and 5-HT3aR, which are expected to be desensitized conformations, 9' is the only constriction in the pore that poses an energetic barrier to ion and water permeation (*Basak et al., 2018a*; *Noviello et al., 2021*; *Polovinkin et al., 2018*; *Zhuang et al., 2022*). It was previously suggested that the agonist-bound structures of ELIC and 5-HT3aR may be pre-active states (*Petroff et al., 2022*; *Felt et al., 2024*), and that the channel may somehow be limited from entering a desensitized state due to detergent or nanodisc conditions (*Dalal et al., 2024*). Based on the liposome structures, we conclude that 9' is the desensitization gate for ELIC and possibly other cation-selective pLGICs. This conclusion is consistent with studies showing that mutation of 9' (L240A) in ELIC appears to eliminate desensitization (*Gonzalez-Gutierrez et al., 2012*).

Lipid-like densities are also resolved in the agonist-bound ELIC5 structure (*Figure 2—figure supplement 3*). The most prominent lipid densities are in the outer TMD consisting of four lipid-like tails. Two

of these tails fill intersubunit (M1–M3) and intrasubunit (M3–M4) grooves and meet to form a putative phospholipid headgroup near R117 of the β6–β7 interfacial loops. A very similar phospholipid density was observed previously in agonist-bound ELIC5 in nanodiscs (e.g. MSP1E3D1) (*Figure 2—figure supplement 3*), and streamlined free energy perturbation calculations determined that this is likely POPG and not POPE or POPC (*Petroff et al., 2022*). Another lipid-like tail adjacent to M1 joins with the density in the intersubunit site, making a three-pronged lipid-like density. The ELIC5 structure in MSP1E3D1 also showed a similar but weaker density. Therefore, the lipid densities observed in the liposome structure of the activated state of ELIC are very similar to those observed in a nanodisc structure, suggesting that ELIC and not the nanodisc is primarily determining the interaction of these phospholipids. Lipid-like densities are not well appreciated in the apo WT ELIC or agonist-bound WT ELIC structures, possibly due to the lower resolution of these structures or differences in the nature of the lipid interactions (*Petroff et al., 2022*).

## ELIC is associated with the MSP even in large nanodiscs

Nanodiscs affect the structure of ELIC possibly through interactions of ELIC with the MSP (*Dalal et al., 2024*; *Nouel Barreto et al., 2025*). Larger nanodiscs may decrease the probability of interactions with the MSP and therefore be superior membrane mimetics (*Nouel Barreto et al., 2025*). To test this possibility, we reconstituted WT ELIC in spNW25, a circularized nanodisc with an average diameter of 25 nm (*Zhang et al., 2021b*), for single particle cryo-EM analysis, and determined a structure of ELIC in the presence of agonist (propylamine). spNW25 nanodiscs are significantly larger than the 11 nm (spMSP1D1) and 15 nm (spNW15) nanodiscs previously used for agonist-bound WT ELIC structures

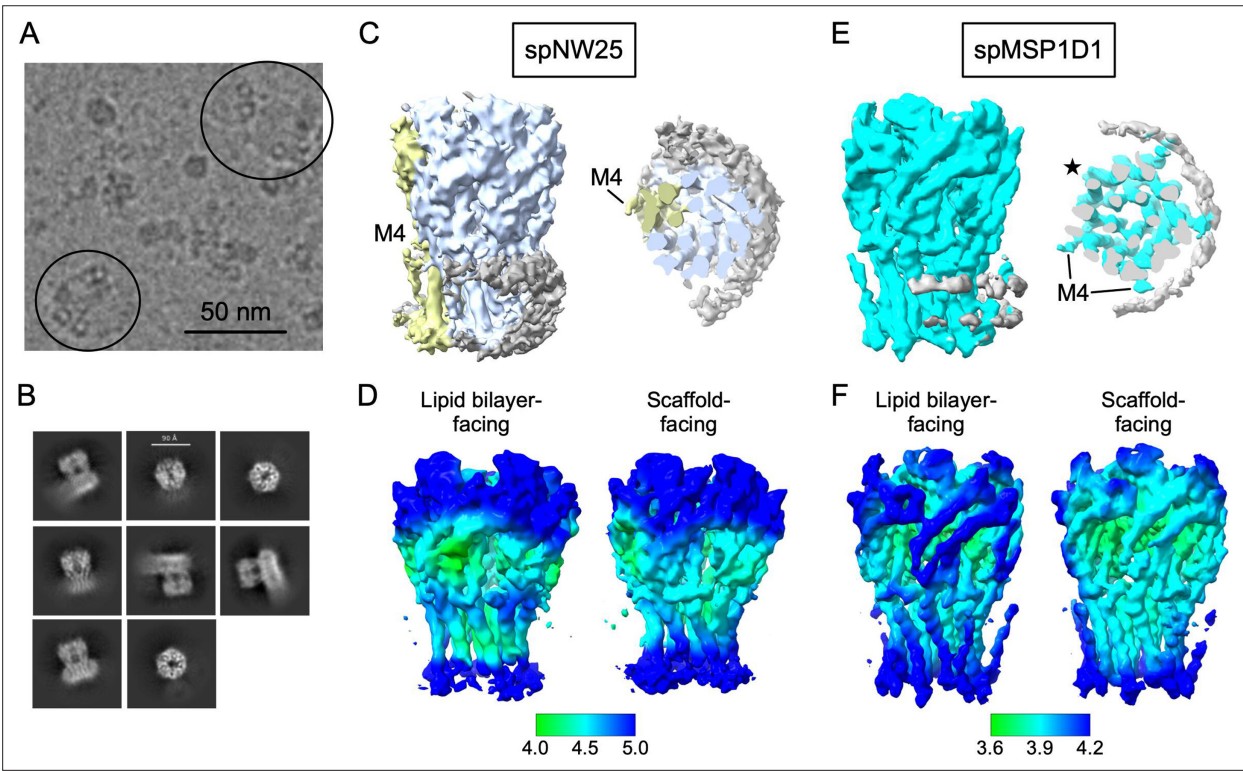

**Figure 3.** Single particle cryo-EM of ELIC in spNW25 nanodiscs. (**A**) Cryo-EM micrograph of WT ELIC with 10 mM propylamine in spNW25 nanodiscs. The circles indicate top-down views of spNW25 nanodiscs with multiple ELIC ion channels at the rim of the nanodisc. (**B**) 2D class averages of WT ELIC in spNW25. (**C**) C1 cryo-EM map of agonist-bound WT ELIC in spNW25 at low contour showing the nanodisc scaffold density (gray). M4 is resolved in one of the ELIC subunits colored yellow. (**D**) Displays of the local resolution for the map in (**C**). Shown are the aspects of ELIC facing the lipid bilayer and the nanodisc scaffold. (**E**) C1 cryo-EM map of agonist-bound WT ELIC in spMSP1D1 at low contour showing the nanodisc scaffold density (gray). M4 densities can be appreciated in all subunits except one of the lipid bilayer-facing subunits (star). (**F**) Displays of the local resolution for the map in (**E**). Shown are the aspects of ELIC facing the lipid bilayer and the nanodisc scaffold.

The online version of this article includes the following figure supplement(s) for figure 3:

**Figure supplement 1.** Summary of single particle cryo-EM analysis of WT ELIC with agonist in spNW25 nanodiscs.

(*Dalal et al., 2024*). Initial inspection of the micrographs showed multiple ELIC particles in most nanodiscs and localization of the particles to the rim of the nanodisc (*Figure 3A*). Consistent with this observation, side views of the 2D class averages (*Figure 3B*) and the resulting 3D reconstruction using C1 symmetry (*Figure 3C*, *Figure 3—figure supplement 1*) showed significant MSP density on one side of the TMD. A recent study showed that MSP densities can be resolved in the structures of many membrane proteins (*Koh et al., 2025*). We therefore re-analyzed the dataset of agonist-bound WT ELIC in spMSP1D1 with C1 symmetry; the resulting map showed strong MSP densities associated with ELIC (*Figure 3E*). Taken together, the structures indicate that ELIC has a strong affinity for the MSP or nanodisc rim irrespective of nanodisc size.

We next examined whether the C1 reconstructions of agonist-bound WT ELIC in spMSP1D1 or spNW25 nanodiscs exhibit any asymmetric features associated with contact with the MSP. In the spMSP1D1, the cryo-EM map showed stronger densities for M4 in subunits adjacent to the MSP, and the local resolutions of those subunits are higher in both the TMD and ECD (*Figure 3E, F*). This agrees with the finding in other nanodisc-reconstituted membrane proteins that interactions with the MSP stabilize the local conformation of the protein (*Koh et al., 2025*). In contrast, the C1 reconstruction of agonist-bound ELIC in spNW25 showed better local resolution in subunits facing the lipid bilayer (*Figure 3D*). In addition, M4 was only resolved in one of the lipid bilayer-facing subunits (*Figure 3C*). A C5 reconstruction of agonist-bound WT ELIC in spNW25 also did not show densities for M4 (*Figure 3—figure supplement 1*). The C1 maps of agonist-bound ELIC in nanodiscs suggest that the MSP can influence the local dynamics of the protein.

The micrographs and 2D class averages of ELIC in spNW25 showed multiple ELIC particles in each nanodisc. ~10% of particles from the 2D class averages showed two adjacent pentamers (*Figure 3—figure supplement 1*), but these dimers of pentamers did not yield high-resolution reconstructions. 2D class averages of dimers of pentamers were never observed in the liposome datasets. This observation suggests that reconstitution of membrane proteins in large nanodiscs may be a strategy to promote the association of multiple membrane proteins. Whether the formation of dimers or pentamers alters the function of ELIC is not known.

## Comparison of ELIC structures in liposomes and nanodiscs

The structures of ELIC in nanodiscs differ from the structures in liposomes to varying degrees. From this and previous studies (*Dalal et al., 2024*; *Petroff et al., 2022*), the following structures of ELIC in nanodiscs have been determined: apo WT ELIC in MSP1E3D1 (PDB 8D65) and spMSP1D1 (PDB 8F35), agonist-bound ELIC5 in MSP1E3D1 (PDB 8VUW) and spNW15 (PDB 8TWV), and agonist-bound WT ELIC in MSP1E3D1 (PDB 8D66), spMSP1D1 (PDB 8F34), spNW15 (PDB 8TWZ), and spNW25 (PDB 9NH4). To examine the effect of nanodisc size on ELIC structure, we compared agonist-bound WT ELIC in liposomes with spMSP1D1 (11 nm), spNW15 (15 nm), and spNW25 (25 nm). Unexpectedly, the structure in the largest nanodisc, spNW25, deviates most from the liposome structure. In the spNW25 structure, there is a shortening of M1 and M2 and tilting of M3 toward the membrane that decreases the height of these transmembrane helices along the pore axis by ~1–1.5 Å compared to the liposome and spMSP1D1 structures (*Figure 4A*, *Supplementary file 2*). Along with this, M2 is tilted toward the pore axis, narrowing the pore even further at L240 (9') and Q233 (2') (*Figure 4B*). In spNW15, M2 tilts toward the pore axis at the intracellular end, slightly narrowing the pore, as was reported previously (*Dalal et al., 2024*). The spMSP1D1 structure most closely matches the liposome structure, except there is ~1 Å outward (away from the pore axis) translation of M4 in the liposome structure compared to the spMSP1D1 structure (*Figure 4C*). There is no M4 density in the C5 maps of agonist-bound WT ELIC in spNW15 and spNW25, suggesting that larger nanodiscs increase the structural heterogeneity of this helix.

In apo WT ELIC, the structures in MSP1E3D1 or spMS1PD1 are not significantly different from the structure in liposomes. In agonist-bound ELIC5, M4 is translated (2 Å) and tilted (~5°) outward (away from the pore axis) in the liposome structure compared to the MSP1E3D1 or spNW15 structures (*Figure 4D*). Along with this, there is a slight outward expansion of the M1, M2, and M3 transmembrane helices away from the pore axis, indicating that the liposome is producing an activated structure that is more loosely packed in the TMD.

We further analyzed the agonist-bound WT ELIC and ELIC5 structures by measuring the area between subunit interfaces in the TMD and the buried surface area between M4 and the rest of the

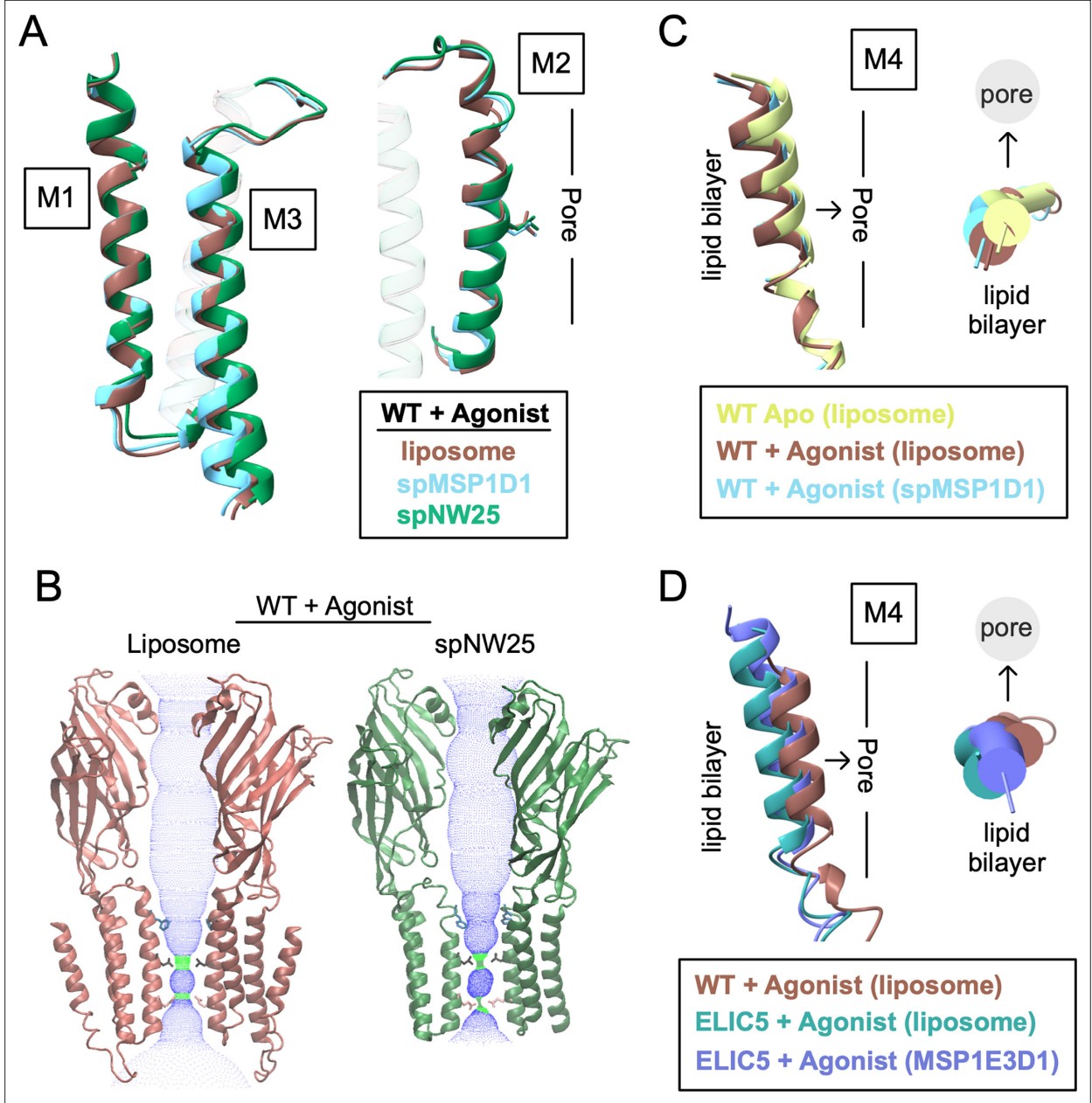

**Figure 4.** Comparison of ELIC structures in liposomes and nanodiscs. (**A**) Comparison of the M1, M2, and M3 helices in structures of WT ELIC with agonist in liposomes, spMSP1D1 and spNW25. Shown in M2 is the side chain of L240 and the approximate location of the ion conduction pore relative to M2. Images are from a global superposition of the structures. (**B**) Two opposing subunits from structures of WT ELIC with agonist in liposomes and spNW25. Shown are dot plots of the pore dimensions generated using HOLE (*Smart et al., 1996*), and the side chains for F247 (16'), L240 (9'), and Q233 (2'). (**C**) Comparison of M4 in WT ELIC in liposomes, WT ELIC with agonist in liposomes, and WT ELIC with agonist in spMSP1D1 showing a side view and top view (from extracellular side) of the helix. The approximate location of the lipid bilayer and ion conduction pore is indicated relative to M4. Images are from a global superposition of the structures. (**D**) Same representation as (**C**) comparing M4 in WT ELIC with agonist in liposomes, ELIC5 with agonist in liposomes, and ELIC5 with agonist in MSP1E3D1.

protein. There is an overall trend where the liposome structures having a smaller area between subunit interfaces in the TMD and a smaller buried surface area between M4 and the rest of the protein (*Supplementary file 3*). This means that the TMD is more loosely packed in the agonist-bound structures in liposomes compared to nanodiscs. Accordingly, the total buried surface area is also consistently smaller in the liposome structures compared to the nanodisc structures (*Supplementary file 3*).

In summary, the agonist-bound structures of WT ELIC or ELIC5 in liposomes deviate from the nano-disc structures, particularly in M4. The structure of the TMD in liposomes is generally more loosely packed than in nanodiscs. This is consistent with MD simulations of ELIC in nanodiscs showing that the outward tilt of M4 is reduced by interactions with the MSP (*Dalal et al., 2024*). Contrary to what was previously suggested (*Dalal et al., 2024*), the agonist-bound WT ELIC structure in a large nanodisc (spNW25) deviates most from the structure in liposomes, indicating that such large nanodiscs do not recapitulate the membrane environment of a liposome.

## Discussion

Single particle cryo-EM of ion channels in liposomes provides the opportunity to study structure and function in the same preparation (*Mandala and MacKinnon, 2022*; *Yang et al., 2022*). There has been uncertainty as to whether agonist-bound structures of ELIC in lipid nanodiscs represent a desensitized state (*Dalal et al., 2024*; *Kumar et al., 2020b*). Using liposomes that support ELIC function, we have greater confidence that the structures presented in this study capture resting, activated, and desensitized conformations. The desensitized conformation shows that the 9' leucine is likely the desensitization gate in ELIC. The ability to obtain structures of ELIC in liposomes sets the stage for time-resolved cryo-EM structural analysis of a pLGIC where the rates of activation and desensitization can be monitored (*Berriman and Unwin, 1994*).

We previously observed variation in agonist-bound structures of ELIC in nanodiscs of different size (*Dalal et al., 2024*). The nanodisc scaffold could affect the structure of ELIC by perturbing the lipid bilayer or by directly interacting with ELIC itself. Therefore, we predicted that larger nanodiscs with a larger lipid bilayer area may better recapitulate the environment of a lipid bilayer that is not surrounded by a nanodisc scaffold. This prediction is not supported by the agonist-bound structure of ELIC in large ~25 nm nanodiscs, which shows the greatest deviation from the liposome structure. Strikingly, ELIC localizes only to the rim of this large nanodisc. This is anticipated to produce a non-uniform environment (i.e. MSP on one side and a lipid bilayer on the other), and the differences in the local resolution of C1 reconstructions of ELIC in nanodiscs may be a consequence of direct interactions with the MSP. It is also likely that the spNW25 nanodisc alters lipid bilayer properties particularly at the rim of the nanodisc, and that this contributes to the observed changes in the TMD structure. There is evidence that other membrane proteins in nanodiscs have an affinity for the MSP (*Koh et al., 2025*; *Kern et al., 2021*), suggesting that the perturbing effects of large nanodiscs on ELIC structure may apply to other membrane proteins.

In pLGICs, M4 is in the periphery of the TMD and is thought to be a lipid sensor due to its contact with the membrane (*Hénault et al., 2015*). In ELIC, the structure and dynamics of M4 influence agonist responses (*Hénault et al., 2019*). Based on the liposome structures, activation of ELIC involves an outward translation and tilting of M4 toward the membrane; this movement of M4 is greatest in the activated state and intermediate in the desensitized state. In both activated and desensitized structures, nanodiscs limit this agonist-induced conformational change in M4, and in desensitized structures, some nanodiscs (i.e. spNW15 and spNW25) appear to increase the conformational heterogeneity of M4. Therefore, we infer from the structural data that nanodiscs affect agonist responses in ELIC, suggesting that structure determination in liposomes is necessary to capture snapshots of activated and desensitized states of this pLGIC.

## Methods

### Expression, purification, and liposome reconstitution of ELIC

ELIC was expressed from pET-26-MBP-ELIC provided by Raimund Dutzler (Addgene plasmid #39239) in OverExpress C43 (DE3) *E. coli* (Lucigen 60446-1) as previously described (*Dietzen et al., 2022*). The ELIC5 mutant (P254G/C300S/V251Y/G319F/I320F) was made using QuikChange and confirmed by Sanger sequencing (Genewiz). Cultures were grown with Terrific Broth (Sigma T0918) using 0.1 mM IPTG (Sigma I6758) for induction. The cells were lysed with an Avestin C5 emulsifier, and membranes were solubilized with 1% DDM (Anatrace D310S). The extracted protein was purified with amylose resin (New England Biolabs E8022L) and eluted with 40 mM maltose in buffer A (10 MM Tris pH 7.5, 100 mM NaCl, 0.02% DDM). The maltose binding protein was removed by overnight digestion with

HRV-3c protease (Thermo Fisher Scientific 88947). Lastly, the protein was purified over a Sephadex 200 Increase 10/300 size exclusion column (Cytiva 28-9909-44) in Buffer A.

Reconstitution of WT ELIC or ELIC5 in liposomes was performed using dialysis. 2:1:1 POPC:POPE:POPG was dried overnight in a vacuum desiccator to remove chloroform, resuspended in 50 mM HEPES pH 7, 150 mM NaCl, 2% cholate at 10 mg/ml, and sonicated to clarity. The lipid solution was diluted to 5 mg/ml to 1% cholate. For each sample, 0.5 mg of lipid (100 µl) was mixed with 250 µg of WT ELIC or ELIC5 at ~7 mg/ml, producing an ELIC(pentamer):lipid molar ratio of ~1:500. For samples with agonist, propylamine was added at this step to a final concentration of 10 mM such that the proteoliposomes will have propylamine in both the extraliposomal and intraliposomal space. The sample was incubated on ice for 1 hr, and then dialyzed (Thermo Scientific 88402-20K MWCO) against 50 mM HEPES pH 7, 150 mM NaCl with or without agonist over 5–7 days with 6× solution changes of 14 ml each. After dialysis, the sample was ~150 µl. The sample was centrifuged at $10,000 \times g$ for 5 min to remove large aggregates and concentrated to ~50–75 µl using an Amicon Ultra-0.5 100 kDa MWCO centrifugal filter (Millipore Sigma UFC910024). The final lipid concentration was ~5 mg/ml, and this was used directly for freezing for cryo-EM analysis.

## Expression and purification of spNW25, and nanodisc reconstitution

spNW25 was expressed and purified from Addgene plasmid #173484 provided by Huan Bao exactly as previously described using Ni-NTA affinity purification, but without size exclusion chromatography (*Zhang et al., 2021b*). Reconstitution of WT ELIC in spNW25 was performed using a liposome destabilization technique as described previously (*Petroff et al., 2022*). 2:1:1 POPC:POPE:POPG in 50 mM HEPES pH 7, 150 mM NaCl at 7.5 mg/ml was freeze–thawed 3× and extruded with a 400-nm filter (Avanti Polar Lipids). These lipids were destabilized by adding 0.4% DDM and incubating at RT for 3 hr. ELIC and spNW25 were added to the sample with a molar ratio of 1:2:600 of ELIC:spNW25:lipid and incubated at RT for 1.5 hr. DDM was removed to form nanodiscs with Biobeads SM-2 Resin (Bio-Rad 1528920) overnight at 4°C. Nanodiscs were further purified over a Sephadex 200 Increase 10/300 column in 50 mM HEPES pH 7 with 150 mM NaCl. The nanodisc fractions were concentrated to ~0.5 mg/ml, and propylamine was added to a final concentration of 50 mM at least 30 min prior to freezing for cryo-EM analysis.

## Cryo-EM sample preparation and imaging

Cryo-EM samples for agonist-bound WT ELIC in spNW25 were prepared as previously described (*Dalal et al., 2024*). 3 µl of sample was applied to Quantifoil R2/2 copper grids, previously negatively glow discharged using a GloQube (Quorum Technologies) for 60 s. The grids were blotted for 2 s in a 95% humidity environment and vitrified in liquid ethane with a Vitrobot Mark IV (Thermo Fisher Scientific). Images were collected on a Glacios 200 kV Cryo-EM equipped with a Falcon 4 Direct Electron Detector (Thermo Fisher Scientific). The single particle cryo-EM data was acquired by counting mode on the Falcon 4 with EPU software (version 3.5.1). The movies were collected with a pixel size of 1.184, and defocus range of –0.8 to –2.4 µm. Each movie consisted of 45 individual frames with a total dose of 46.6 electrons per Å².

For the proteoliposome samples, 3 µl of sample was applied to a Quantifoil R2/2 gold grid and incubated for 3 min in a 95% humidity environment. The grid was then blotted for 2 s, followed by a second sample application. After the second sample application, the grid was incubated at 95% humidity for 30 s, blotted for 2 s, and vitrified in liquid ethane. Images were collected on a Krios 300 kV Cryo-EM equipped with a Falcon 4 Direct Electron Detector. Single particle cryo-EM data was acquired using counting mode on the Falcon 4 with the EPU software 3.8.1. For apo WT ELIC and agonist-bound ELIC5, movies were collected with a pixel size of 0.868, and for agonist-bound WT ELIC, movies were collected with a pixel size of 0.865. A total dose of 55.4 electrons per Å² was used for apo WT ELIC, 54.4 electrons per Å² for agonist-bound WT ELIC, and 56.9 electrons per Å² for agonist-bound ELIC5.

## Single particle analysis and model building

Single particle cryo-EM analysis of WT ELIC in spNW25 with 50 mM propylamine was performed using CryoSPARC v4.6.0 (*Punjani et al., 2017*) and Relion5.0. Initial processing was performed with CryoSPARC including Patch Motion Correction, Patch CTF, and particle picking using templates from

a previously determined ELIC map. Extracted particles were cleaned up with multiple rounds of 2D classification. To increase the number of picked particles, Topaz particle picking was performed by separately training with top/down, tilted, and side views of ELIC from the last 2D classification job, and these particles underwent multiple rounds of 2D classification. The resulting ~787,500 particles underwent multiple heterogeneous refinement jobs using C1 symmetry. For the C1 reconstruction, the best class from the final heterogeneous refinement job was processed by non-uniform refinement in CryoSPARC. For the C5 reconstruction, the ~787,500 particles underwent multiple heterogeneous refinement jobs in CryoSPARC using C5 symmetry, followed by multiple 3D classifications in Relion 5.0 with blush regularization and C5 symmetry. The final post-processed map was generated in Relion 5.0.

Each of the three proteoliposome samples was analyzed in a similar fashion as the approach used for the ELIC spNW25 structure, and the workflow is highlighted in *Figure 1—figure supplement 1*, *Figure 1—figure supplement 2*, and *Figure 1—figure supplement 3*. Movies were motion corrected and CTF estimation was performed in cryoSPARC. Particle picking was performed with blob picker, which yielded poor results, but many rounds of 2D classification eventually yielded a small number of particles (several thousand) producing class averages that resembled top/down, tilted, and membrane-embedded side views. These particles were used for Topaz training, and the extracted particles were further processed by 2D classification. The resulting 2D class averages were improved and, again, used for Topaz training separating top/down and tilted from side views. This was performed two to three times for each dataset. When there was no significant increase in the number of good particles from Topaz picking, an ab initio reconstruction was performed to generate initial volumes for heterogeneous refinement. It was apparent from the side views of the 2D class averages that ELIC was oriented inward and outward in the liposomes. Accordingly, ab initio reconstructions produced volumes with opposite membrane curvature indicative of inward- and outward-facing channels. Multiple rounds of heterogeneous refinement with C1 and C5 symmetry were performed with the inward- and outward-facing volumes as well as two to three junk volumes. The C1 refinement jobs did not produce 3D reconstructions of sufficient quality for further refinement. Therefore, we proceeded with refinement and 3D classification using C5 symmetry. The resulting particles were used for non-uniform refinement and local refinement with a mask for the entire ELIC or ELIC5 protein. The local refinement job generally produced better maps, probably because it masked out the liposome membrane from the refinement procedure. In general, better maps could be obtained with further processing in Relion 5.0, except for agonist-bound ELIC5, which yielded a final outward-facing map of 2.8 Å resolution. Particles from inward- and outward-facing classes underwent multiple rounds of 3D classification in Relion 5.0 without and with a mask for the entire ELIC protein. The masked 3D classification jobs yielded the best final classes for post-process refinement.

For the apo WT ELIC structures in liposomes, an initial model of ELIC was obtained from PDB 8F35. For all agonist-bound WT structures, an initial model was obtained from PDB 8F34. For the agonist-bound ELIC5 structures in liposomes, an initial model was obtained from PDB 8VUW. These were used to perform real space refinement in PHENIX 1.19.2 (*Adams et al., 2010*). The structure of ELIC was manually built into the cryo-EM density map using COOT 0.9.6 (*Emsley et al., 2010*), refined in PHENIX, and this process was repeated iteratively. Propylamine was fit into the density in the agonist binding site based on the orientation predicted from MD simulations (*Kumar et al., 2020b*). For all C5 models, a single subunit was built, refined, and propagated to the other subunits. 3D volume visualization, analysis, and image preparation were performed using PyMOL 2.5.2 and ChimeraX 1.6.1.

## Fluorescence stopped-flow liposomal flux assay

The fluorescence stopped-flow liposomal flux measurements were performed on the SX20 stopped-flow spectrofluorometer (Applied Photophysics) as previously described (*Petroff et al., 2022*), but with some modifications to assay ELIC activity in liposomes formed by dialysis—the same liposome reconstitution method that was used for cryo-EM. 2:1:1 POPC:POPE:POPG lipids were dried and reconstituted in 10 mM HEPES pH 7, 100 mM $NaNO_3$, 1% cholate at 5 mg/ml and sonicated to clarity. For each sample, 2.5 mg of lipid (0.5 ml) was used and mixed with 12.5 µg of WT ELIC or 2.5 µg of ELIC5, incubated for 1 hr at 4°C, and dialyzed for 5 days against 14 ml of buffer (50 mM HEPES pH 7, 100 mM $NaNO_3$) with 6 solution changes at 4°C. Next, the proteoliposome samples were mixed with 1.5 ml of 50 mM HEPES pH 7, 100 mM $NaNO_3$, and 75 mM 8-aminonaphthalene-1,3,6-trisulfonic acid (ANTS, Thermo Fisher Scientific). The samples underwent a single freeze–thaw to introduce ANTS into

the intra-liposomal space. To remove the extra-liposomal ANTS, the samples were diluted to 2.5 ml and run through a PD-10 desalting column (Cytiva) pre-equilibrated with 50 mM HEPES pH 7, 140 mM $NaNO_3$ (Stopped-flow Buffer A), and eluted with 3 ml of this buffer. Lastly, the samples were diluted threefold with Stopped-flow Buffer A.

Two experiments were performed to assess activation and desensitization of WT ELIC, and activation of ELIC5. For WT ELIC, activation and desensitization proceed in the millisecond and second time scales, respectively, so a rapid sequential-mixing experiment was performed with delay times from 10 ms to 60 s. WT ELIC proteoliposomes were mixed 1:1 with Stopped-flow Buffer A containing 2× concentration of propylamine (20 mM). After a variable delay time, the sample was mixed 1:1 with quenching buffer (50 mM HEPES pH 7, 90 mM $NaNO_3$, 50 mM $TlNO_3$), and ANTS fluorescence was recorded for 1 s. Multiple repeat measurements were obtained for each delay time and averaged. For ELIC5, activation proceeds in the seconds time scale and the channel is anticipated to not desensitize. To test whether ELIC5 exhibits any desensitization over a long time scale (up to 20 min) we modified the assay to add propylamine manually. At predetermined times (1–20 min), a single mixing experiment was performed that involved 1:1 mixing with quenching buffer followed by measurement of ANTS fluorescence for 1 s. The data were analyzed using the Pro-Data SX software from Applied Photophysics. The fluorescence quenching traces were fit to a stretched exponential, and the rate of $Tl^+$ influx was determined at 2 ms (*Posson et al., 2018*).

## Acknowledgements

This study was supported by grants R35GM137957 to WWC, AHA postdoctoral fellowship and the Owen's Anesthesiology Research Fellowship to VD, and the Foundation for Anesthesia Education and Research Mentored Research Training Grant to BKT.

## Additional information

### Funding

| Funder | Grant reference number | Author |
| --- | --- | --- |
| National Institute of General Medical Sciences | R35GM137957 | Wayland WL Cheng |
| American Heart Association | 10.58275/AHA. 24POST1189869.pc.gr. 190934 | Vikram Dalal |
| Foundation for Anesthesia Education and Research | Mentored Research Training Grant | Brandon K Tan |
| University of Washington | Owen's Anesthesiology Research Fellowship | Vikram Dalal |

The funders had no role in study design, data collection, and interpretation, or the decision to submit the work for publication.

### Author contributions

Vikram Dalal, Brandon K Tan, Data curation, Formal analysis, Writing - review and editing; Hanrui Xu, Data curation, Writing - review and editing; Wayland WL Cheng, Conceptualization, Data curation, Formal analysis, Supervision, Funding acquisition, Investigation, Methodology, Writing - original draft

### Author ORCIDs

Wayland WL Cheng ⓘ https://orcid.org/0000-0002-9529-9820

Reviewer #1 (Public review): https://doi.org/10.7554/eLife.106728.3.sa1
Reviewer #2 (Public review): https://doi.org/10.7554/eLife.106728.3.sa2
Author response https://doi.org/10.7554/eLife.106728.3.sa3

# Additional files

## Supplementary files

Supplementary file 1. Summary of cryo-EM data collection and refinement parameters.

Supplementary file 2. Distance along the pore axis (i.e. perpendicular to the lipid membrane) between the indicated residues (Cα atoms) for each structure. The measurements show changes in the height of each transmembrane helix along the pore axis.

Supplementary file 3. The area at the interface between subunits in the transmembrane domain (TMD) and the buried surface area. The area at the interface between subunits was determined using the PDBePISA server (*Krissinel and Henrick, 2007*), and the buried surface area between M4 and the rest of the ELIC protein was determined using ChimeraX1.6.1.

MDAR checklist

## Data availability

The data supporting the findings of this study are available within the paper and supplementary information files. The cryo-EM maps have been deposited in the Electron Microscopy Data Bank (EMDB) under accession codes EMD-49382, EMD-49383, EMD-49384, EMD-49385, EMD-49390, EMD-49391, EMD-49392, and EMD-49400. The structural coordinates have been deposited in the RCSB Protein Data Bank (PDB) under the accession codes 9NGC, 9NGF, 9NGG, 9NGI, 9NGQ, 9NGR, 9NGS, and 9NH4.

The following datasets were generated:

| Author(s) | Year | Dataset title | Dataset URL | Database and Identifier |
|---|---|---|---|---|
| Dalal V, Cheng WWL | 2025 | ELIC5 with propylamine facing ECD outwards in liposomes with 2:1:1 POPC:POPE:POPG | https://www.ebi.ac.uk/emdb/EMD-49382 | EMDataBank, EMD-49382 |
| Dalal V, Cheng WWL | 2025 | ELIC facing ECD outwards in liposomes with 2:1:1 POPC:POPE:POPG | https://www.ebi.ac.uk/emdb/EMD-49383 | EMDataBank, EMD-49383 |
| Dalal V, Cheng WWL | 2025 | ELIC facing ECD inwards in liposomes with 2:1:1 POPC:POPE:POPG | https://www.ebi.ac.uk/emdb/EMD-49384 | EMDataBank, EMD-49384 |
| Dalal V, Cheng WWL | 2025 | ELIC5 with propylamine facing ECD inwards in liposomes with 2:1:1 POPC:POPE:POPG | https://www.ebi.ac.uk/emdb/EMD-49385 | EMDataBank, EMD-49385 |
| Dalal V, Cheng WWL | 2025 | ELIC state 1 with propylamine facing ECD outwards in liposomes with 2:1:1 POPC:POPE:POPG | https://www.ebi.ac.uk/emdb/EMD-49390 | EMDataBank, EMD-49390 |
| Dalal V, Cheng WWL | 2025 | ELIC state 2 with propylamine facing ECD outwards in liposomes with 2:1:1 POPC:POPE:POPG | https://www.ebi.ac.uk/emdb/EMD-49391 | EMDataBank, EMD-49391 |
| Dalal V, Cheng WWL | 2025 | ELIC with propylamine facing ECD inwards in liposomes with 2:1:1 POPC:POPE:POPG | https://www.ebi.ac.uk/emdb/EMD-49392 | EMDataBank, EMD-49392 |
| Dalal V, Cheng WWL | 2025 | ELIC with propylamine in spNW25 nanodiscs with 2:1:1 POPC:POPE:POPG | https://www.ebi.ac.uk/emdb/EMD-49400 | EMDataBank, EMD-49400 |
| Dalal V, Cheng WWL | 2025 | ELIC5 with propylamine facing ECD outwards in liposomes with 2:1:1 POPC:POPE:POPG | https://doi.org/10.2210/pdb9ngc/pdb | Worldwide Protein Data Bank, 10.2210/pdb9ngc/pdb |

*Continued on next page*

*Continued*

| Author(s) | Year | Dataset title | Dataset URL | Database and Identifier |
|---|---|---|---|---|
| Dalal V, Cheng WWL | 2025 | ELIC facing ECD outwards in liposomes with 2:1:1 POPC:POPE:POPG | https://doi.org/10.2210/pdb9ngf/pdb | Worldwide Protein Data Bank, 10.2210/pdb9ngf/pdb |
| Dalal V, Cheng WWL | 2025 | ELIC facing ECD inwards in liposomes with 2:1:1 POPC:POPE:POPG | https://doi.org/10.2210/pdb9ngg/pdb | Worldwide Protein Data Bank, 10.2210/pdb9ngg/pdb |
| Dalal V, Cheng WWL | 2025 | ELIC5 with propylamine facing ECD inwards in liposomes with 2:1:1 POPC:POPE:POPG | https://doi.org/10.2210/pdb9ngi/pdb | Worldwide Protein Data Bank, 10.2210/pdb9ngi/pdb |
| Dalal V, Cheng WWL | 2025 | ELIC state 1 with propylamine facing ECD outwards in liposomes with 2:1:1 POPC:POPE:POPG | https://doi.org/10.2210/pdb9ngq/pdb | Worldwide Protein Data Bank, 10.2210/pdb9ngq/pdb |
| Dalal V, Cheng WWL | 2025 | ELIC state 2 with propylamine facing ECD outwards in liposomes with 2:1:1 POPC:POPE:POPG | https://doi.org/10.2210/pdb9ngr/pdb | Worldwide Protein Data Bank, 10.2210/pdb9ngr/pdb |
| Dalal V, Cheng WWL | 2025 | ELIC with propylamine facing ECD inwards in liposomes with 2:1:1 POPC:POPE:POPG | https://doi.org/10.2210/pdb9ngs/pdb | Worldwide Protein Data Bank, 10.2210/pdb9ngs/pdb |
| Dalal V, Cheng WWL | 2025 | ELIC with propylamine in spNW25 nanodiscs with 2:1:1 POPC:POPE:POPG | https://doi.org/10.2210/pdb9nh4/pdb | Worldwide Protein Data Bank, 10.2210/pdb9nh4/pdb |

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
