## [Editor Report · eLife Assessment]

The reported cryo-EM imaging of a pentameric ligand-gated ion channel in liposomes as opposed to nanodiscs has both broad implications and contributes **valuable** methodological advances to the structural investigation of membrane receptors. The comparison of structures assigned to distinct functional states in liposomes versus nanodiscs is **convincing** and will aid membrane protein structural biologists in selection of functionally relevant membrane reconstitution environments.

---

## [Referee Report · Reviewer #1 (Public review)]

Summary:

The authors, Dalal, et. al., determined cryo-EM structures of open, closed, and desensitized states of the pentameric ligand-gated ion channel ELIC reconstituted in liposomes, and compared them to structures determined in varying nanodisc diameters. They argue that the liposomal reconstitution method is more representative of functional ELIC channels, as they were able to test and recapitulate channel kinetics through stopped-flow thallium flux liposomal assay. The authors and others have described channel interactions with membrane scaffold proteins (MSP), initially thought to be in a size-dependent manner. However, the authors reported their cryo-EM ELIC structure interacts with the large nanodisc spNW25, contrary to their original hypotheses. This suggests that the channels interactions with MSPs might alter its structure, possibly influencing the functional states of the channel. Thus, the authors describe reconstitution in liposomes are more representative of the native structure and can recapitulate all channel states.

Strengths:

Cryo-EM structural determination from proteoliposomes is promising methodology within the ion channel field due to their large surface area and lack of MSP or other membrane memetics that could alter channel structure. The authors succeeded in comparing structures determined in liposomes to those in a wide range of nanodisc diameters. This comparison gives rise to important discussions for other membrane protein structural studies when deciding the best method for individual circumstances.

Weaknesses:

As the overarching goal of the study was to determine structural differences of ELIC in detergent nanodiscs and liposomes. The authors stated they determined open, closed, and desensitized states of ELIC reconstituted in liposomes and suggest the desensitization gate is at the 9' region of the pore. However, limited functional data was provided when determining the functional states of the channel with most of the evidence deriving from structures, which only provides snapshots of channels.

---

## [Referee Report · Reviewer #2 (Public review)]

Summary

The report by Dalas and colleagues introduces a significant novelty in the field of pentameric ligand-gated ion channels (pLGICs). Within this family of receptors, numerous structures are available, but a widely recognised problem remains in assigning structures to functional states observed in biological membranes. Here, the authors obtain both structural and functional information of a pLGIC in a liposome environment. The model receptor ELIC is captured in the resting, desensitised and open states. Structures in large nanodiscs, possibly biased by receptor-scaffold protein interactions, are also reported. Altogether these results set the stage for the adoption of liposomes as a proxy for the biological membranes, for cryoEM studies of pLGICs and membrane proteins in general.

Strengths

The structural data is comprehensive, with structures in liposomes in the 3 main states (and for each, both inward-facing and outward-facing), and an agonist-bound structure in the large spNW25 nanodisc (and a retreatment of previous data obtained in a smaller disc). It adds up to a series of work from the same team that constitutes a much-needed exploration of various types of environment for the transmembrane domain of pLGICs. The structural analysis is thorough.

The tone of the report is particularly pleasant, in the sense that the authors' claims are not inflated. For instance, a sentence such as "By performing structural and functional characterization under the same reconstitution conditions, we increase our confidence in the functional annotation of these structures." is exemplary.

Weakness

All the details necessary to reproduce the work are present in the Methods. Nevertheless, the biochemistry might have been shown and discussed in greater details. While I do believe that liposomes will be in most cases better than, say, nanodiscs, the process that leads from the protein in its membrane down to the liposome will play a big role in preserving the native structure.

---

## [Author Response]

The following is the authors’ response to the original reviews

**Public Reviews:**

**Reviewer #1 (Public review):**
Summary:The authors, Dalal, et. al., determined cryo-EM structures of open, closed, and desensitized states of the pentameric ligand-gated ion channel ELIC reconstituted in liposomes, and compared them to structures determined in varying nanodisc diameters. They argue that the liposomal reconstitution method is more representative of functional ELIC channels, as they were able to test and recapitulate channel kinetics through stopped-flow thallium flux liposomal assay. The authors and others have described channel interactions with membrane scaffold proteins (MSP), initially thought to be in a size-dependent manner. However, the authors reported that their cryo-EM ELIC structure interacts with the large nanodisc spNW25, contrary to their original hypotheses. This suggests that the channel's interactions with MSPs might alter its structure, possibly not accurately representing/reflecting functional states of the channel.Strengths:Cryo-EM structural determination from proteoliposomes is a promising methodology within the ion channel field due to their large surface area and lack of MSP or other membrane mimetics that could alter channel structure. Comparing liposomal ELIC to structures in various-sized nanodiscs gives rise to important discussions for other membrane protein structural studies when deciding the best method for individual circumstances.Weaknesses:The overarching goal of the study was to determine structural differences of ELIC in detergent nanodiscs and liposomes. Including comparisons of the results to the native bacterial lipid environment would provide a more encompassing discussion of how the determined liposome structures might or might not relate to the native receptor in its native environment. The authors stated they determined open, closed, and desensitized states of ELIC reconstituted in liposomes and suggest the desensitization gate is at the 9' region of the pore. However, no functional studies were performed to validate this statement.

The goal of this study was to determine structures of ELIC in the same lipid environment in which its function is characterized. However, it is also worth noting that phosphatidylethanolamine and phosphatidylglyerol, two lipids used for the liposome formation, are necessary for ELIC function (PMID 36385237) and principal lipid components of gram-negative bacterial membranes in which ELIC is expressed.

The desensitized structure of ELIC in liposomes shows a pore diameter at the hydrophobic L240 (9’) residue of 3.3 Å, which is anticipated to pose a large energetic barrier to the passage of ions due to the hydrophobic effect. We have included a graphical representation of pore diameters from the HOLE analysis for all liposome structures in Supplementary Figure 6B. While we have not tested the role of L240 in desensitization with functional experiments, it was shown by Gonzalez-Gutierrez and colleagues (PMID 22474383) that the L240A mutation apparently eliminates desensitization in ELIC. This finding is consistent with L240 (9’) being the desensitization gate of ELIC. We have referenced this study when discussing the desensitization gate in the Results.

**Reviewer #2 (Public review):**
SummaryThe report by Dalas and colleagues introduces a significant novelty in the field of pentameric ligand-gated ion channels (pLGICs). Within this family of receptors, numerous structures are available, but a widely recognised problem remains in assigning structures to functional states observed in biological membranes. Here, the authors obtain both structural and functional information of a pLGIC in a liposome environment. The model receptor ELIC is captured in the resting, desensitized, and open states. Structures in large nanodiscs, possibly biased by receptor-scaffold protein interactions, are also reported. Altogether, these results set the stage for the adoption of liposomes as a proxy for the biological membranes, for cryoEM studies of pLGICs and membrane proteins in general.StrengthsThe structural data is comprehensive, with structures in liposomes in the 3 main states (and for each, both inward-facing and outward-facing), and an agonist-bound structure in the large spNW25 nanodisc (and a retreatment of previous data obtained in a smaller disc). It adds up to a series of work from the same team that constitutes a much-needed exploration of various types of environment for the transmembrane domain of pLGICs. The structural analysis is thorough.The tone of the report is particularly pleasant, in the sense that the authors' claims are not inflated. For instance, a sentence such as "By performing structural and functional characterization under the same reconstitution conditions, we increase our confidence in the functional annotation of these structures." is exemplary.WeaknessesCore parts of the method are not described and/or discussed in enough detail. While I do believe that liposomes will be, in most cases, better than, say, nanodiscs, the process that leads from the protein in its membrane down to the liposome will play a big role in preserving the native structure, and should be an integral part of the report. Therefore, I strongly felt that biochemistry should be better described and discussed. The results section starts with "Optimal reconstitution of ELIC in liposomes [...] was achieved by dialysis". There is no information on why dialysis is optimal, what it was compared to, the distribution of liposome sizes using different preparation techniques, etc... Reading the title, I would have expected a couple of paragraphs and figure panels on liposome reconstitution. Similarly, potential biochemical challenges are not discussed. The methods section mentions that the sample was "dialyzed [...] over 5-7 days". In such a time window, most of the members of this protein family would aggregate, and it is therefore a protocol that can not be directly generalised. This has to be mentioned explicitly, and a discussion on why this can't be done in two days, what else the authors tested (biobeads? ... ?) would strengthen the manuscript.To a lesser extent, the relative lack of both technical details and of a broad discussion also pertains to the cryoEM and thallium flux results. Regarding the cryoEM part, the authors focus their analysis on reconstructions from outward-facing particles on the basis of their better resolutions, yet there was little discussion about it. Is it common for liposome-based structures? Are inward-facing reconstructions worse because of the increased background due to electrons going through two membranes? Are there often impurities inside the liposomes (we see some in the figures)? The influence of the membrane mimetics on conformation could be discussed by referring to other families of proteins where it has been explored (for instance, ABC transporters, but I'm sure there are many other examples). If there are studies in other families of channels in liposomes that were inspirational, those could be mentioned. Regarding thallium flux assays, one argument is that they give access to kinetics and set the stage for time-resolved cryoEM, but if I did not miss it, no comparison of kinetics with other techniques, such as electrophysiology, nor references to eventual pioneer time-resolved studies are provided.Altogether, in my view, an updated version would benefit from insisting on every aspect of the methodological development. I may well be wrong, but I see this paper more like a milestone on sample prep for cryoEM imaging than being about the details of the ELIC conformations.

Additions have been made to the Results and Discussion sections elaborating on the following points: (1) reconstitution of ELIC in liposomes using dialysis, the advantage of this over other methods such as biobeads, and whether the dialysis protocol can be shortened for other less stable proteins; (2) the issue of separating outward- and inward-facing channels; (3) referencing the effect of nanodiscs on ABC transporters, structures of membrane proteins in liposomes, and pioneering time-resolved cryo-EM studies; and (4) comparison of the kinetics of ELIC gating kinetics with electrophysiology measurements. With regards to the first point, it should be noted that all necessary details are provided in the Methods to reproduce the experiments including the reconstitution and stopped-flow thallium flux assay. It is also important to note that the same preparation for making proteoliposomes was used for assessing function using the stopped-flow thallium flux assay and for determining the structure by cryo-EM. This is now stated in the Results.

**Recommendations for the authors:**

**Reviewer #1 (Recommendations for the authors):**
Major revisions:(1) The authors suggest that the desensitization gate is located at the 9' region within the pore. However, as stated by the authors, the 2' residues function as the desensitization gate in related channels. In a few of their HOLE analyzed structures (e.g. Figure 2B and 4B), there seems to be a constriction also at 2', but this finding is not discussed in the context of desensitization. Further functional testing of mutated 9' and/or 2' gates would bolster the argument for the location of the desensitization gate.

As stated above, we have included HOLE plots of pore radius in Supplementary Fig. 6B and referenced the study showing that the L240A mutation (9’) in ELIC (PMID 22474383) appears to eliminate desensitization. This result along with the narrow pore diameter at 9’ in the desensitized structure suggests that 9’ is likely a desensitization gate in ELIC. In contrast, mutation of Q233 (2’) to a cysteine in a previous study produced a channel that still desensitizes (PMID 25960405). Since Q233 is a hydrophilic residue in contrast to L240, Q233 probably does not pose the same energetic barrier to ion translocation as L240 based on the structure.

(2) In discussing functional states of ELIC and ELIC5 in different reconstitution methods, the authors reference constriction sites determined by HOLE analysis software. These constriction sites were key evidence for the authors to determine functional state, however, it is difficult to discern pore sizes based on the figures. Pore diameters and clear color designation (ie, green vs orange) with the figures would greatly aid their discussions.

HOLE plots are displayed in Supplementary Fig. 6B and pore diameters are not provided in the text.

(3) The authors had an intriguing finding that ELIC dimers are found in spNW25 scaffolds. Is there any functional evidence to suggest they could be functioning as dimers?

There is no evidence that the function of ELIC or other pLGICs is altered by the formation of dimers of pentamers. Therefore, while this result is intriguing and likely facilitated by concentrating multiple ELIC pentamers within the nanodisc, it is not clear if these interactions have any functional importance. We have stated this in the Results.

(4) Thallium flux assay to validate channel function within proteoliposomes. Proteoliposomes are known to be generally very leaky membranes, would be good to have controls without ELIC added to determine baseline changes in fluorescence.

We have established from multiple previous studies that liposomes composed of 2:1:1 POPC:POPE:POPG (PMID 36385237 and 31724949) do not show significant thallium flux as measured by the stopped-flow assay (PMID 29058195) in the absence of ELIC activity. Furthermore, in the present study, the data in Fig. 1A of WT ELIC shows a low thallium flux rate 60 seconds after exposure to agonist when the ion channel has mostly desensitized. Therefore, this data serves also as a control indicating that the high thallium flux rates in response to agonist (at earlier delay times) are not due to leak, but rather due to ELIC channel activity.

Minor revisions:(1) Abstract and introduction. 'Liganded' should be ligand

We removed this word and changed it to “agonist-bound” for consistency throughout the manuscript.

(2) Inconsistent formatting of FSC graphs in Supplemental Figure 4

The difference is a consequence of the different formatting between cryoSPARC and Relion FSC graphs.

**Reviewer #2 (Recommendations for the authors):**
Minor writing remarks:The present report builds on previous work from the same team, and to my eye it would be a plus if this were conveyed more explicitly. I see it as a strength to explore various developments in several papers that complement each other. E.g in the introduction when citing reference 12 (Dalal 2024), later in introducing ref 15 (Petroff 2022), I wish I was reminded of the main findings and how they fit with the new results.

We have expanded on the Results and Discussion detailing key findings from these studies that are relevant to the current study.

Suggestions for analysis:Data treatment. Maybe I missed it, but I wondered if C1 vs C5 treatment of the liposome data showed any interesting differences? When I think about the biological membrane, I picture it as a very crowded place with lots of neighbouring proteins. I would not be surprised if, similarly to what they do in discs, the receptor would tend to stick to, or bump into, anything present also in liposomes (a neighboring liposome, some undefined density inside the liposome).

We attempted to perform C1 heterogeneous refinement jobs in cryoSPARC and C1 3D classification in Relion5. For the WT datasets, these did not produce 3D reconstructions that were of sufficient quality for further refinement. For ELIC5 with agonist, the C1 reconstructions were not different than the C5 reconstructions. Furthermore, there was no evidence of dimers of pentamers from the 2D or 3D treatments, unlike what was observed in the spNW25 nanodiscs. This is likely because the density of ELIC pentamers in the liposomes was too low to capture these transient interactions. We have included this information in the Methods.

In data treatment, we sometimes find only what we're looking for. I wondered if the authors tried to find, for instance, the open and D conformations in the resting dataset during classifications.

This is an interesting question since some population of ELIC channels could visit a desensitized conformation in the absence of agonist and this would not be detected in our flux assay. After extensive heterogeneous refinement jobs in cryoSPARC and 3D classification jobs in Relion5, we did not detect any unexpected structures such as open/desensitized conformations in the apo dataset.

In the analysis of the M4 motions, is there info to be gained by looking at how it interacts with the rest of the TMD? For instance, I wondered if the buried surface area between M4 and the rest was changed. Also one could imagine to look at that M4 separately in outward-facing and inward-facing conformations (because the tension due to the bilayer will not be the same in the outer layer in both orientations - intuitively, I'd expect different levels of M4 motions)

We have expanded our analysis of the structures as recommended. We determined the buried surface area between M4 and the rest of the channel in the liganded WT and ELIC5 structures in liposomes and nanodiscs, as well as the area between the TMD interfaces for these structures. There appears to be a pattern where liposome structures show less buried surface area between M4 and the rest of the channel, and less area at the TMD interfaces. Overall, this suggests that the liposome structures of ELIC in the open-channel or desensitized conformations are more loosely packed in the TMD compared to the nanodisc structures.

We have also further discussed the issue of separating outward- and inward-facing conformations in the Results. The problem with classifying outward- and inward-facing orientations is that top/down or tilted views of the particles cannot be easily distinguished as coming from channels in one orientation or the other, unless there are conformational differences between outward- and inward-facing channels that would allow for their separation during 3D heterogeneous refinement or 3D classification. Furthermore, since the inward-facing reconstructions are of much lower resolution than the outward-facing reconstructions, we suspect that these particles are more heterogeneous possibly containing junk, multiple conformations, or particles that are both inward- and outward-facing. On the other hand, the outward-facing structures are of good quality, and therefore we are more confident that these come from a more homogeneous set of particles that are likely outward-facing (Note that most particles are outward facing based on side views of the 2D class averages). That said, when examining the conformation of M4 in outward- and inward-facing structures, we do not see any significant differences with the caveat that the inward-facing structures are of poor quality and that inward- and outward-facing particles may not have been well-separated.